# Effect of myopia on the progression of normal tension glaucoma

Chun-Mei Hsueh[1,2]*, Jong-Shiuan Yeh[3,4,5], Jau-Der Ho[1,2]

1 Department of Ophthalmology, Taipei Medical University Hospital, Taipei Medical University, Taipei, Taiwan, 2 Department of Ophthalmology, School of Medicine, College of Medicine, Taipei Medical University, Taipei, Taiwan, 3 Division of Cardiology, Department of Internal Medicine, Wan Fang Hospital, Taipei Medical University, Taipei, Taiwan, 4 Taipei Heart Institute, Taipei Medical University, Taipei, Taiwan, 5 Division of Cardiology, Department of Internal Medicine, School of Medicine, College of Medicine, Taipei Medical University, Taipei, Taiwan

* crismay9143@gmail.com

## Abstract

### Purpose

Identify risk factors of progression in treated normal-tension glaucoma (NTG) in highly myopic and non-highly myopic eyes.

### Methods

This retrospective, observational case series study included 42 highly myopic glaucoma (HMG, <-6D) eyes and 39 non-highly myopic glaucoma (NHG,≧-6D) eyes. Glaucoma progression was determined by serial visual field data. Univariate and multivariate logistic regression method were used to detect associations between potential risk factors and glaucoma progression.

### Results

Among 81 eyes from 81 normal-tension glaucoma patients (mean follow-up, 3.10 years), 20 of 42 eye (45.24%) in the HMG and 14 of 39 eyes (35.90%) in the NHG showed progression. The HMG group had larger optic disc tilt ratio (p = 0.007) and thinner inferior macular thickness (P = 0.03) than the NHG group. Changes in the linear regression values for MD for each group were as follows: -0.652 dB/year for the HMG and -0.717 dB/year for the NHG (P = 0.298). Basal pattern standard deviation (PSD) (OR: 1.55, p = 0.016) and post treatment IOP (OR = 1.54, p = 0.043) were risk factors for visual field progression in normal tension glaucoma patients. In subgroup analysis of HMG patients, PSD (OR: 2.77, p = 0.017) was a risk factor for visual field progression.

### Conclusion

Reduction IOP was postulated to be contributing in the prevention of visual field progression, especially in highly myopic NTG patients with large basal pattern standard deviation.

**Data Availability Statement:** All relevant data are within the manuscript and its Supporting information files.

**Funding:** The authors received no specific funding for this work.

**Competing interests:** The authors have declared that no competing interests exist.

## Introduction

There is a high prevalence of myopia, 22.9% of the world population had myopia and 163 million people with high myopia (2.7% of the world population). In east Asia, approximately one fifth of the myopic population has high myopia (≦-6 diopters), which is a common cause of visual impairment worldwide [1]. In a recent meta-analysis study which comprised 24 studies, the pooled odds ratio association with open-angle glaucoma (OAG) was 1.88 for any myopia degree and 4.1 for high myopia, with a cutoff value of -6D. For each unit (1-D) increase in myopia, the risk of glaucoma increases by approximately 20% [2]. More severe myopia was associated with greater odds of glaucoma [3–6], but the levels of myopia associated with glaucoma progression is still a subject of debate [7–11].

The POAG prevalence in Asian populations is between 1.0% and 3.9% and normal tension glaucoma (NTG) comprises the majority (46.9–92%) of open-angle glaucoma in Asian epidemiologic studies, whereas the calculated mean proportion of NTG was 33.7% in white population [12]. Intraocular ocular pressure(IOP)-independent risk factors, such as ocular structural (myopia [5]) and systemic vascular factors (migraine, hypertension, hypotension, diabetes mellitus [13]) have presented associations with NTG. IOP is an important risk factor for NTG progression [14], but glaucoma progression still develop in 37–59% of treated NTG patients during follow up [15–18]. Myopic NTG have characteristic structural change, like tilt and torsion, and atypical retinal nerve fiber layer defect, especially in highly myopic eyes with early glaucoma [19]. This study aimed to evaluate the risk factors of progression in treated NTG between highly myopic and non-highly myopic eyes.

## Materials and methods

In this retrospective, observational case series study, 42 highly myopic normal tension glaucoma (HMG, <-6D) eyes and 39 non-highly myopic normal tension glaucoma (NHG, ≧-6D) eyes treated from Jan. 2013 to Jun. 2021 were studied. Refractive status was measured via a manifest refraction test using a spherical equivalent. Non-high myopia group include emmetropia and hypermetropia if it's open angle and normal tension glaucoma. All participants received a complete ophthalmic examination which included best-corrected visual acuity, non-contact tonometry (TONOREF III, NIDEK, Japan) and iCare tonometer (TA01i), gonioscopy, central corneal thickness with specular microscope (EM 3000, Tomey, USA), slit-lamp biomicroscopy, stereoscopic disc photography (Canon. CR-2 AF, Japan); and a VF test (Kowa AP-5000c, Japan). Peripapillary RNFL thickness was measured using Spectralis SD-OCT (Heidelberg Engineering, Germany). SD-OCT scans acquire a total of 1536 A-scan points from a 3.45-mm circle centered on the optic disc. Images with non-centered scans or signal strength 15 or less were excluded. For assessment of total macular layer (TML) thickness asymmetry, the posterior pole asymmetry analysis (PPAA) scan was performed. The superior and inferior hemisphere thickness were automatically generated and the macular asymmetry which was assessed by TML thickness differences between the superior and inferior hemispheres were calculated from the PPAA report.

Patient inclusion criteria were all participants at least 18 years old, best-corrected visual acuity of 20/40 or better, eyes had open angles on gonioscopic examinations and normal tension glaucoma was confirmed by optic nerve damage and VF defect. We excluded patients with history of secondary glaucoma (eg, steroid use, trauma, intraocular tumor) or other diseases known to affect the VF (eg, pituitary lesions, Alzheimer disease, stroke, and diabetic retinopathy), inability to perform perimetry reliably, or life-threatening disease. If both eyes of the same patient were found to be eligible, one eye was randomly selected for analysis. Patients were followed up every three months for at least two years. VF and OCT examinations were

performed twice at 3-month intervals for baseline data and then 6-month intervals follow-up. Only reliable VF test results (false-positive errors <15%, false-negative errors<15%, and fixation loss <20%) were included in the analysis. VF progression was defined as a significant (P<0.01) sensitivity loss >1dB/year detected in≧2 adjacent test locations in the same fields [20].

All patients received only medical treatment. When NTG was diagnosed, the patients received Prostaglandin analogue. During the follow-up, if glaucoma progressed or patient can't tolerate the side effects of Prostaglandin analogue, brimonidine 2mg/ml (Allergan, Inc, Irvine, California, USA) or Brinzolamide 1% (Alcon, Swiss) usually was added or replaced. The study design followed the principles of the Declaration of Helsinki and the study was approved by the Ethics Committee of the Taipei Medical University (TMU-N202203184), which also acts on behalf of its affiliated hospital, Taipei Medical University Hospital.

Tilt ratio was measured as the longest-to-shortest diameter ratio of the optic disc. The angle (in degrees) between the vertical meridian and the disc's long axis was defined as the torsional angle. Blood pressure was measured with a digital automatic blood pressure monitor. Mean arterial pressure (MAP) was calculated as 1/3 mean SBP + 2/3 mean DBP. Mean ocular perfusion pressure (MOPP) was calculated as 2/3 MAP-mean IOP. Systolic perfusion pressure (SOPP) was calculated as the mean SBP-mean IOP, and diastolic perfusion pressure (DOPP) was calculated as the mean DBP-mean IOP.

## Statistical analysis

To calculate the sample size, a pilot study was conducted, which considered VF and RNFL thickness data from both the high myopia and no-high myopia groups, using the baseline and the end of the study as time points for the calculation. Successively, the differences between the time points for the variable (VF and RNFL thickness) in the no-high myopia and high myopia groups were calculated. Through this procedure, two sample t test for independent groups was estimated using the pooled standard deviation for VF and RNFL thickness (±3.58 and ±11.1, respectively). After establishing the pooled standard deviation, clinically meaningful differences of d = -2.9 and 4.2um was hypothesized for VF and RNFL thickness, respectively. Finally, use SPSS sample power3 to calculate the sample sizes, with $\alpha$ = 0.05 and power = 80% were applied, and a final sample size of n = 80 (no-high myopia = 40 and high myopia = 40) were estimated.

Independent t test was used to compare continuous variables between HMG and NHG. The chi-square test was used to compare categorical data. Changes of MD per year for each group were performed with linear regression analysis. Using univariate and multivariate logistic regression analyses, the study assessed risk factors of NTG progression and subgroup analysis for HMG and NHG were performed. Covariates such as age, gender, DM, HTN, baseline IOP, post-treatment IOP, baseline MD, baseline PSD were controlled in the multivariate analysis. Risk factors associated with VF progression in patients with NTG without diabetes mellitus or hypertension were also analyzed with multivariate logistic regression.

The study used the SPSS (version 26.0; SPSS Inc, Chicago, IL, USA) to perform these analyses. Differences between the two groups were considered significant for *p* values of <0.05.

## Results

Among 81 eyes from 81 normal-tension glaucoma patients (mean follow-up, 3.1 years), 19 of 42 eye (45.24%) in the HMG and 14 of 39 eyes (35.90%) in the NHG showed progression (P = 0.39). The HMG group was younger (47.4 ± 8.19 years) than the NHG group (56.54 ± 10.8 years, P = 0.07). Tilt ratio was larger in the HMG group than in the NMG group

(P = 0.007). Total macular thickness (P = 0.05) and inferior macular thickness (P = 0.03) were significantly thinner in the HMG group. The patient characteristics are shown in Table 1. Changes in the linear regression values for MD for each group were as follows: -0.652 dB/year for the HMG and -0.717 dB/year for the NHG (P = 0.298). Pattern standard deviation (PSD) (OR: 1.55, p = 0.016) and post treatment IOP (OR = 1.54, p = 0.043) were risk factors for visual field progression in normal tension glaucoma patients (Table 2). In our subgroup analysis for HMG patients, progression group was male dominant (P<0.001) (Table 3) and PSD (OR: 2.77, p = 0.017) was a risk factor for visual field progression. But no such significant association in non-high myopic NTG patients. The highly-myopic NTG patients evidenced VF progression of 1.37-fold that per 1-D increased myopia (p = 0.388) (Table 4). Subgroup analysis was also performed for patients with NTG without DM or HTN, and PSD was a risk factor for VF progression (OR = 2.03, P = 0.017) (Table 5).

**Table 1. Characteristic of participants.**

|  | HMG (n = 42) | NHG (n = 39) | P- value |
|---|---|---|---|
| Spherical equivalent(D) | -8.37 ± 1.51 | -3.30 ± 1.99 | **<0.001** |
| Age | 47.40 ± 8.19 | 56.54 ± 10.80 | 0.07 |
| Gender (male: female) | 20:22 | 22:17 | 0.43 |
| Family history (n, %) | 9 (21.43) | 2 (5.13) | **0.03** |
| Diabetes mellitus (n, %) | 7(16.67) | 6(15.38) | 0.88 |
| Migraine (n, %) | 6 (14.29) | 10 (25.64) | 0.20 |
| Hyperlipidemia (n, %) | 12(28.57) | 9(23.08) | 0.57 |
| Hypertension (n, %) | 10 (23.81) | 9(23.08) | 0.94 |
| Arrythmia (n, %) | 2(4.76) | 6(15.38) | 0.11 |
| Baseline IOP (mmHg) | 17.49 ± 2.05 | 17.65 ± 2.30 | 0.75 |
| Post-treatment IOP (mmHg) | 14.03 ± 2.01 | 13.28 ± 2.18 | 0.12 |
| Systolic BP | 120.79±15.70 | 123.22±16.89 | 0.51 |
| Diastolic BP | 75.79±10.09 | 77.11±13.75 | 0.63 |
| MOPP | 44.85±8.30 | 43.92±9.49 | 0.65 |
| CCT (um) | 546.90 ± 30.49 | 540.88 ± 40.21 | 0.47 |
| Torsion degree | -6.90 ± 3.70 | -6.04 ± 5.14 | 0.39 |
| Tilt ratio | 1.48 ± 0.57 | 1.21 ± 0.18 | **0.007** |
| Baseline VF MD (dB) | -5.29 ± 3.65 | -5.33 ± 2.60 | 0.95 |
| Baseline PSD (dB) | 4.81 ± 3.20 | 4.52 ± 2.43 | 0.67 |
| Baseline RNFL (um) | 75.05 ± 9.74 | 76.13 ± 12.27 | 0.67 |
| Final MD (dB) | -6.90 ± 3.79 | -6.85 ± 3.49 | 0.96 |
| Final PSD (dB) | 4.59 ± 2.90 | 4.31 ± 2.38 | 0.70 |
| Final RNFLT (um) | 69.47 ± 10.47 | 73.31 ± 12.47 | 0.14 |
| TMT (um) | 268.50±13.24 | 274.53±10.88 | 0.05 |
| Asymmetry (um) | 19.77±8.97 | 17.97±8.83 | 0.44 |
| Superior TMT (um) | 278.30±12.37 | 283.16±13.64 | 0.15 |
| Inferior TMT (um) | 258.53±15.35 | 265.77±10.05 | **0.03** |
| Drug number (PG %) | 1.49±0.82 (66.7) | 1.46±0.74 (76.9) | 0.87 |
| Progression/Non progression | 19(45.24%)/23 | 14 (35.90%) /25 | 0.39 |
| Following Time (year) | 3.12 ± 1.08 | 2.94 ± 0.94 | 0.42 |

HMG indicates highly myopic glaucoma; NHG, non-highly myopic glaucoma; CCT, central corneal thickness; IOP, intraocular pressure; BP, blood pressure; MOPP, mean ocular perfusion pressure; VF MD, visual field mean deviation; PSD, pattern standard deviation; RNFLT, retinal nerve fiber layer thickness; TMT, total macular layer thickness; Asymmetry, superior and inferior total macular layer thickness difference; PG, prostaglandin analogue

**Table 2. Prediction of progression in Myopic Normal-Tension Glaucoma.**

| Total (n = 81) | Univariate Analysis | | | Multivariate Analysis | | |
|---|---|---|---|---|---|---|
| Factors | Odds Ratio | (95% CI) | P-value[a] | Odds Ratio | (95% CI) | P- value[a] |
| Age, years | 1.01 | (0.97–1.05) | 0.721 | | | |
| Myopia (1-D increase) | 1.00 | (0.87–1.16) | 0.977 | | | |
| Gender | 0.29 | (0.11–0.77) | 0.009 | | | |
| Diabetes mellitus | 0.24 | (0.07–0.87) | 0.03 | | | |
| Hypertension | 0.40 | (0.14–1.14) | 0.087 | | | |
| Baseline IOP | 1.17 | (0.94–1.47) | 0.16 | | | |
| MD (1dB increase) | 0.90 | (0.77–1.05) | 0.165 | 1.30 | (0.94–1.79) | 0.110 |
| PSD (1dB increase) | 1.25 | (1.03–1.50) | 0.022 | 1.55 | (1.09–2.22) | **0.016** |
| Post-treatment IOP (1-mm Hg increase) | 1.20 | (0.96–1.50) | 0.108 | 1.54 | (1.01–2.35) | **0.043** |

IOP indicated intraocular pressure; MD, mean deviation; PSD, pattern standard deviation; CI, confidence interval; TMT, total macular layer thickness; STMT: superior macular layer; ITMT: inferior macular layer; MOPP, mean ocular perfusion pressure

[a]P values were calculated using logistic regression model

## Discussion

In a study compared the visual field defect among primary angle-closure glaucoma, high-tension glaucoma and normal-tension glaucoma, the superior hemifield was affected more severely than the inferior hemifield in all three subtypes of primary glaucoma. This asymmetric tendency was more pronounced in NTG [21]. Myopic eye usually had optic nerve head deformity and peripapillary atrophy. Because the RNFL is thickest at the peripapillary retina, and

**Table 3. Characteristics of HMG progression in Normal-Tension Glaucoma.**

| | Progression (n = 19) | Non-progression (n = 23) | P- value |
|---|---|---|---|
| Age | 47.68 ± 7.54 | 47.17 ± 8.86 | 0.84 |
| Spherical equivalent(D) | -8.20±1.37 | -8.51±1.63 | 0.51 |
| Gender (male:female) | 14: 5 | 6: 17 | **0.002** |
| Baseline IOP | 18.00 ± 1.91 | 17.09 ±2.11 | 0.16 |
| Post-treatment IOP | 14.56 ± 1.76 | 13.59 ± 2.13 | 0.13 |
| VF MD (dB) | -5.77±4.28 | -4.85±3.01 | 0.45 |
| VF PSD(dB) | 5.72±3.63 | 4.13±2.73 | 0.15 |
| Final VF MD (dB) | -8.46±4.41 | -5.65±2.72 | **0.025** |
| Initial RNFL (*um*) | 73.72±10.89 | 76.14±8.81 | 0.44 |
| Final RNFL (*um*) | 65.63±10.51 | 72.95±9.37 | **0.025** |
| Final VF PSD (dB) | 5.78±3.70 | 3.68±1.71 | **0.048** |
| TMT(*um*) | 263.25±11.98 | 274.50±12.35 | **0.017** |
| Asymmetry (*um*) | 20.56±9.30 | 18.86±8.84 | 0.61 |
| STMT (*um*) | 273.44±11.33 | 283.86±11.44 | **0.018** |
| ITMT (*um*) | 252.88±14.13 | 265.00±14.52 | **0.028** |
| Final T RNFLT(*um*) | 67.16±20.78 | 79.81±17.99 | **0.046** |
| Final TI RNFLT(*um*) | 63.42±28.45 | 80.81±25.62 | **0.049** |
| Drug number (PG %) | 2.00±1.155 (78.9) | 1.19±0.402(56.5) | **0.005** |

IOP, intraocular pressure; VF MD, visual field mean deviation; PSD, pattern standard deviation; RNFLT, retinal nerve fiber layer thickness; TMT, total macular layer thickness; Asymmetry, superior and inferior total macular layer thickness difference; STMT: superior total macular layer thickness; ITMT: inferior total macular layer thickness; TI: temporal-inferior; T: temporal; PG, prostaglandin analogue

**Table 4. Multivariate risk factors associated with visual field progression in patients with highly myopic and non-highly Myopic Normal-Tension Glaucoma.**

| | HMG (n = 42) | | | NHG (n = 39) | | |
|---|---|---|---|---|---|---|
| | Odds Ratio | (95% CI) | P- value | Odds Ratio | (95% CI) | P- value |
| Age, years | 1.04 | (0.84–1.28) | 0.752 | 1.02 | (0.91–1.15) | 0.705 |
| Myopia (1-D increase) | 1.37 | (0.67–2.79) | 0.388 | 1.16 | (0.66–2.04) | 0.597 |
| Baseline IOP (1-mmHg increase) | 1.28 | (0.67–2.48) | 0.456 | 0.83 | (0.50–1.38) | 0.470 |
| MD (1dB increase) | 2.06 | (1.01–4.18) | 0.046 | 1.34 | (0.75–2.44) | 0.311 |
| PSD (1dB increase) | 2.77 | (1.20–6.39) | **0.017** | 1.34 | (0.76–2.33) | 0.310 |
| Post-treatment IOP (1-mm Hg increase) | 1.64 | (0.83–3.25) | 0.155 | 1.74 | (0.84–3.62) | 0.138 |

CI, confidence interval; IOP indicated intraocular pressure; MD, mean deviation; PSD, pattern standard deviation

[a]P values were calculated using logistic regression model

50% of the retinal ganglion cell is located within the macula [22], combined measurements of the structure changes and structure asymmetry in circmpapillary RNFL and macular area may be helpful in diagnosing glaucoma in high myopic eyes [23]. There was a strong correlation between RNFL defects and retinal thinning in the macula and asymmetry between the superior and inferior macula, correlation with a larger PSD [24]. In our study, PSD was the risk factor for NTG progression, especially in the highly myopic NTG (OR: 2.77, P = 0.017) In the European Glaucoma Prevention Study (EGPS) and Ocular Hypertension Treatment Study (OHTS), higher PSD was a factor that predicted the development of open angle glaucoma (HR: 1.66 in EGPS; HR: 1.27 in OHTS) [25].

Temporal inferior RNFL is important in detecting progression, and the temporal RNFL, corresponding to the papillomacular bundle, is relatively well preserved along the course of glaucoma progression [26]. But high myopia and NTG eyes seem to have atypical RNFL defects. According to Kimura and associates [19], highly myopic eyes are more susceptible to papillomacular bundle damage in early glaucoma. One possible mechanism would be the mechanical stress from scleral stretching at the temporal area of lamina cribrosa associated with myopic eyeball elongation. Progressive tilting of the optic disc with nasal shift of temporal optic disc margin and concurrent development of peripapillary atrophy occurred in children with myopic shift [27]. Longer axial length, larger optic disc and NTG are risk factors for papillomacular bundle defects [28]. Our results showed that high myopic NTG patients have thinner final temporal and temporal-inferior RNFL thickness compared to non-highly myopic patients. Kim and associates [29] reported that RNFL defects were wider and closer to the macula in the high myopia compared with the low to moderate myopia and emmetropia group in

**Table 5. Multivariate risk factors associated with visual field progression in Normal-Tension Glaucoma patients without diabetes mellitus or hypertension.**

| n = 57 | Odds Ratio | (95%CI) | P-value |
|---|---|---|---|
| Myopia (1-D increase) | 1.06 | (0.74–1.54) | 0.748 |
| Baselined IOP (1-mm Hg increase) | 1.46 | (0.86–2.50) | 0.165 |
| MD (dB) | 1.67 | (1.03–2.70) | 0.038 |
| PSD (1dB increase) | 2.03 | (1.14–3.63) | **0.017** |
| Post-IOP (mm Hg) | 1.44 | (0.85–2.44) | 0.171 |

CI, confidence interval; IOP indicated intraocular pressure; MD, mean deviation; PSD, pattern standard deviation

[a]P values were calculated using logistic regression model

NTG patients. High myopia group was younger (43.8±11.7 years) than low to moderate myopia group (48.1±10.5 years), and emmetropia group (55.8±10.2 years). Our result is similar. Glaucomatous damage develops at a younger age and more severe in patients with high myopia may be due to more aggressive pathological changes in the posterior pole of the high myopic eye.

The highly-myopic NTG patients evidenced VF progression of 1.37-fold that per 1-D increased myopia (p = 0.388) and 1.16-fold in non-highly myopic NTG patients (p = 0.597). But high myopia (less than -6D) was not a risk factor for glaucoma progression in this study. Several studies analyzed the risk factors of glaucoma progression showed the controversial results [7, 11, 30, 31]. E Chihara et al. reported that high mean intraocular pressure (p = 0.007) and large refractive error (< or = -4 diopters) (p = 0.023) were significant risk factors for visual field loss in patients with primary open-angle glaucoma [31]. But Naito T. et al. reported eyes with VF progression had significantly higher baseline refraction in primary open-angle glaucoma (-1.9±3.8 diopter [D] vs -3.5±3.4 D, P = 0.0048) [11].

In our study, 45.24% in the HMG and 35.90% in the NHG showed progression. The Early Manifest Glaucoma Trial (EMGT) reported progression in 45% treatment group and the Collaborative Normal-Tension Glaucoma Study (CNTGS) reported progression in 12% treatment patients [32]. The reasons of higher progression rate in this study than CNTGS may due to different refractive errors and IOP lowered percentage. Refractive errors in the treated group in CNTGS is -1.09±3.3 D, and our non-high myopic group is -3.30 ± 1.99 D and high myopic group is -8.37 ± 1.51 D. The treated group in CNTGS have intraocular pressure lowered by 30% from baseline (16.9 ±2.1 to 10.6±2.7 mmHg), and our study have IOP lowered by 19.8% in HMG and 24.8% in NHG. Tezel et al. reported that diagnosed of NTG as independent risk factors for progression. 38% of eyes with primary open-angle glaucoma suffered progressive cupping at a mean IOP of 16.9mmHg, and 51% of eyes with NTG suffered progression at a mean IOP of 15mmHg [33]. Post-treatment IOP (OR: 1.54, P = 0.043) was associated with NTG progression in our study. Collaborative Normal-Tension Glaucoma Study [14] showed the importance of lowering IOP to slow down the progression rate in NTG patients. Lee JY et al. [7] reported that 32.5% in the mild to moderate myopic glaucoma and 20.0% in the highly myopic glaucoma showed progression. These results may be interpreted as a lower progression detection rate because of the difficulty in detecting changes in the optic disc/RNFL in HMG, or as a consequence of some of highly myopic eyes that may not be true cases of glaucoma. Studies reported the progression rate about treated normal tension glaucoma were about 45–59.7% in about 5–6 years follow-up [15, 17, 34].

In our study, changes in the linear regression values for MD were -0.652 dB/year for the HMG and -0.717 dB/year for the NHG (P = 0.298). This occurred more rapidly than in the EMGT treatment group (-0.36dB /year) [35] and CNTGS (-0.499 dB/year) [32]. SW Sohn et al. [9] reported changes in the linear regression values for MD were -0.912 dB /year for high myopic groups and -1.113dB for moderate myopia (-3 to -5.99D). High rate of progression (52%) was noted even after treatment. Higher basal PSD and lower ratio of IOP lowering rate than our study may explain the difference.

It is often a challenge to diagnose glaucoma by peripapillary RNFL in myopic eyes due to optic disc deformity. Structural asymmetry parameters performed well, identifying early POAG as well as RNFL thickness [36]. AROCs for thickness measurements tended to increase with increasing glaucoma severity (pre-perimetric, 0.746–0.808; early, 0.842–0.940; advanced, 0.943–0.995), whereas AROCs for asymmetry indices did not have distinct ranges according to glaucoma severity (pre-perimetric, 0.773–0.994; early, 0.861–0.998; advanced, 0.819–0.996) [37]. NTG patients with high myopia had retinal thickness asymmetry in peripapillary RNFL and total macular layer. Asymmetry analysis of retinal thickness can be an adjunctive tool for

the early detection of highly-myopic NTG [23]. In our study, thinner total, superior and inferior macular layer thickness were noted in HMG progression group.

This study was subject to several limitations. First, our sample series of 81 patients was relatively small. Future studies with larger sample size are warranted. Second, this study is retrospective design, we cannot exclude the possibility of selection bias. Third, we use refractive error, not the axial length, for distinguish high myopia from non-high myopia, with a cutoff value of -6D. Axial length measurement may be a more direct assessment of the degree of myopia, as myopia can be induced by lens changes or corneal astigmatism. But eyes with visually significant lens changes and high corneal astigmatism were excluded from our analysis, so those effects might not be great. Fourth, patients with diabetes mellitus and hypertension (HTN) showed thinner peripapillary retinal nerve fiber layer (pRNFL). Diabetes duration and HTN were significant factors affecting the pRNFL thickness [38]. The Low-Pressure Glaucoma Treatment Study reported that use of antihypertension medications was a risk factor for visual field progression [39]. More glaucoma patients with diabetes or HTN need to be analyzed in the future.

## Conclusions

In conclusion, reduction IOP was postulated to be contributing in the prevention of visual field progression, especially in highly myopic NTG patients with large basal pattern standard deviation. Identifying risk factors for glaucoma, particularly normal tension glaucoma with high myopia, could improve disease detection and treatment initiation at earlier stages.

## Supporting information

**S1 Data set.**
(XLSX)

## Author Contributions

**Data curation:** Chun-Mei Hsueh.

**Software:** Jong-Shiuan Yeh.

**Supervision:** Jau-Der Ho.

**Writing – original draft:** Chun-Mei Hsueh.

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
