## [Decision Letter · Decision Letter 0]

17 Apr 2023

PONE-D-23-00521Effect of Myopia on the Progression of Normal Tension Glaucoma: A Retrospective Cohort StudyPLOS ONE

Dear Dr. Hsueh,

Thank you for submitting your manuscript to PLOS ONE. After careful consideration, we feel that it has merit but does not fully meet PLOS ONE’s publication criteria as it currently stands. Therefore, we invite you to submit a revised version of the manuscript that addresses the points raised during the review process.

ACADEMIC EDITOR:Thank you for this elaborate study. The manuscript is generally well written and the study design is robust. However, as noted by Reviewer #1 this idea and subject have been published in previous reports. Please highlight in the Discussion Section the additive information provided by this manuscript that adds to the already published reports.

We look forward to receiving your revised manuscript.

Kind regards,

Nader Hussien Lotfy Bayoumi, M.D., FRCS (Glasgow)

Academic Editor

PLOS ONE

Journal Requirements:

Reviewers' comments:

Reviewer's Responses to Questions

**Comments to the Author**

1. Is the manuscript technically sound, and do the data support the conclusions?

Reviewer #1: Partly

Reviewer #2: Yes

2. Has the statistical analysis been performed appropriately and rigorously? 

Reviewer #1: No

Reviewer #2: Yes

3. Have the authors made all data underlying the findings in their manuscript fully available?

Reviewer #1: Yes

Reviewer #2: Yes

4. Is the manuscript presented in an intelligible fashion and written in standard English?

Reviewer #1: Yes

Reviewer #2: Yes

5. Review Comments to the Author

Reviewer #1: Study Title: “Effect of Myopia on the Progression of Normal Tension Glaucoma: A Retrospective

Cohort Study.”

Summary Review:

This is a retrospective non-interventional cohort study intended to the potential effect of the degree if myopia on the progression of NTG.

Analysis:

A. The study question and idea: the idea is not novel being previously addressed in literature with nearly the same design reaching the same conclusion

“Sohn SW, Song JS, Kee C. Influence of the extent of myopia on the progression of normal-tension glaucoma. Am J Ophthalmol. 2010 May;149(5):831-8”

“Ernest PJ, Schouten JS, Beckers HJ, Hendrikse F, Prins MH, Webers CA. An evidence-based review of prognostic factors for glaucomatous visual field progression. Ophthalmology. 2013 Mar;120(3):512-519“

B. The study design: “A retrospective cohort study”.

C. Points to consider:

1) Choosing patients with a wide significant difference in age (P<0.001) pose a source of bias even if statistically corrected particularly in relatively small samples in addition to concomitant presence of vascular diseases like diabetes for instance which per se could lead to decrease in the RNFLZ thickness

2) The authors did not mention the number of medications or the type of medications needed to control IOP and what about the effect of these drugs on RNFL or visual field.

3) Also was the effect of systemic treatment for controlling Diabetes mellitus or Hypertension on RNFL thickness evaluated??

“Lee MW, Park GS, Lim HB, Lee WH, Kim MS, Lee YH, Kim JY. Effect of Systemic Hypertension on Peripapillary RNFL Thickness in Patients with Diabetes without Diabetic Retinopathy. Diabetes. 2021 Nov; 70(11):2663-2667.”

Reviewer #2: The purpose of the study:

(Evaluate the risk factors of progression in treated normal-tension glaucoma (NTG) between highly myopic and non-highly myopic eyes.)

• Change (evaluate) to identify or detect risk factors…….change (between) to (in)

• Define in a clear manner (non-highly myopic eyes):

• Is it mild to moderate myopia as it appears from the spherical equivalent in this group of your results?

• The expression (non-high myopia) is not well-defined from another aspect as the term may include emmetropia and hypermetropia also.

Methods:

• The sentence (Among 81 eyes from 73 normal tension glaucoma patients ..) is in conflict with the other sentence you mentioned (If both eyes of the same patient were found to be eligible, one eye was randomly selected for analysis.)

Please clarify this point regarding the number of eyes included in the study and if “both eyes of the same patient” were included in the study in some situations.

• Please define the type of tonometers you used in the study.

• You included treated patients in the study, but you did not mention the type of treatment they received either medical, laser, or surgical which may affect the progression of glaucoma. Please explain this point.

Results:

• Regarding the baseline data in both groups, the age in the HMG group was significantly less (45.95 ± 9.04 years vs 57.35 ± 10.85 years, p < 0.001)

Did the significant difference in age affect the calculation of risk factors?

• The multivariate Cox proportional hazards model indicated that myopia (HR: 1.2, P=0.019) was risk factor for normal tension glaucoma progression.

• Also in conclusions (Myopia was risk factor for glaucoma progression in treated normal tension glaucoma patients).

Please add (a) before (risk factor)

Which degree of myopia you mean?

6. PLOS authors have the option to publish the peer review history of their article (what does this mean?). If published, this will include your full peer review and any attached files.

Reviewer #1: No

Reviewer #2: No

---

## [Author Response · Author response to Decision Letter 0]

4 Jun 2023

The purpose of the study:

(Identify the risk factors of progression in treated normal-tension glaucoma (NTG) in highly myopic and non-highly myopic eyes.) 

• Change (evaluate) to identify or detect risk factors…….change (between) to (in)

1. Define in a clear manner (non-highly myopic eyes): include emmetropia and hypermetropia if it’s open angle and normal tension glaucoma 

• Is it mild to moderate myopia as it appears from the spherical equivalent in this group of your results? Yes

• The expression (non-high myopia) is not well-defined from another aspect as the term may include emmetropia and hypermetropia also.

Methods: 

2. The sentence (Among 81 eyes from 73 normal tension glaucoma patients ..) is in conflict with the other sentence you mentioned (If both eyes of the same patient were found to be eligible, one eye was randomly selected for analysis.) Please clarify this point regarding the number of eyes included in the study and if “both eyes of the same patient” were included in the study in some situations.

 After revision, we change to “Among 81 eyes from 81 normal-tension glaucoma patients”, so it’s not conflict with” If both eyes of the same patient were found to be eligible, one eye was randomly selected for analysis.” 

3. 

• Please define the type of tonometers you used in the study. 

Non-contact tonometer ((TONOREF III, NIDEK, Japan) and i-Care tonometer (TA01i)

• You included treated patients in the study, but you did not mention the type of treatment they received either medical, laser, or surgical which may affect the progression of glaucoma. Please explain this point.

All patients received only medical treatment, and we analyzed the drug numbers, the percentage of prostaglandin analogue and the percentage of intraocular pressure decrease after treatment. When NTG was diagnosed, the patients received Prostaglandin analogue. During the follow-up, if glaucoma progressed or patient can’t tolerate the side effects of Prostaglandin analogue, brimonidine 2mg/ml (Allergan, Inc, Irvine, California, USA) usually was added or replaced. 

Results:

• Regarding the baseline data in both groups, the age in the HMG group was significantly less (45.95 ± 9.04 years vs 57.35 ± 10.85 years, p < 0.001)

Did the significant difference in age affect the calculation of risk factors?

 After the revision, the age difference was not significant. (P=0.07)

• The multivariate Cox proportional hazards model indicated that myopia (HR: 1.2, P=0.019) was risk factor for normal tension glaucoma progression.

• Also in conclusions (Myopia was risk factor for glaucoma progression in treated normal tension glaucoma patients).

Please add (a) before (risk factor)

 Which degree of myopia you mean? 

After revision, basal pattern standard deviation (PSD) (OR: 1.55, p=0.016) and post treatment IOP (OR=1.54, p=0.043) were risk factors for visual field progression in normal tension glaucoma patients. The highly-myopic NTG patients evidenced VF progression of 1.37-fold that per 1-D increased myopia. (p=0.388)

---

## [Editor Report · Decision Letter 1]

12 Jun 2023

Effect of Myopia on the Progression of Normal Tension Glaucoma

PONE-D-23-00521R1

Dear Dr. Hsueh,

We’re pleased to inform you that your manuscript has been judged scientifically suitable for publication and will be formally accepted for publication once it meets all outstanding technical requirements.

Kind regards,

Nader Hussien Lotfy Bayoumi, M.D., FRCS (Glasgow)

Academic Editor

PLOS ONE

Additional Editor Comments (optional):

The authors have addressed all reviewers' comments and concerns. Although the article may be a recapitulation of previous research in the same field, the study is well designed and the conclusions are robust.
---

## [Editor Report · Acceptance letter]

16 Jun 2023

PONE-D-23-00521R1 

Effect of Myopia on the Progression of Normal Tension Glaucoma 

Dear Dr. Hsueh:

I'm pleased to inform you that your manuscript has been deemed suitable for publication in PLOS ONE. Congratulations! Your manuscript is now with our production department. 

Kind regards, 

on behalf of

Professor Nader Hussien Lotfy Bayoumi 

Academic Editor

PLOS ONE